# Significance of Post-Traumatic Growth and Mental Health for Coping in Multiple Sclerosis Caregivers

**DOI:** 10.3390/healthcare11101390

**Published:** 2023-05-11

**Authors:** Irene Gil-González, María Ángeles Pérez-San-Gregorio, Jesús Funuyet-Salas, Rupert Conrad, Agustín Martín-Rodríguez

**Affiliations:** 1Department of Personality, Assessment, and Psychological Treatment, University of Seville, 41018 Seville, Spain; 2Department of Psychosomatic Medicine and Psychotherapy, University Hospital Muenster, 48149 Muenster, Germany

**Keywords:** multiple sclerosis, caregivers, post-traumatic growth, mental health, coping strategies

## Abstract

We investigated the influence of post-traumatic growth (PTG) and mental health (MH) on multiple sclerosis (MS) caregivers’ uses of coping strategies and identified biopsychosocial predictors of proactive or reactive coping. The Short Form Health Survey (SF-12), General Health Questionnaire (GHQ-28), Post-Traumatic Growth Inventory (PGI-21), Brief COPE Questionnaire (COPE-28), and Multidimensional Scale of Perceived Social Support (MSPSS) were used to evaluate 209 caregivers. Higher PTG was related to greater use of emotional support, positive reframing, religion, active coping, instrumental support, planning, denial, self-distraction, self-blaming, and venting. Better MH was associated with greater use of acceptance, while behavioral disengagement and self-distraction were associated with poorer MH. The PTG dimensions relating to others and new possibilities, SF-12 dimensions of physical and emotional roles as well as partnership, not living with the patient, and significant others’ social support were predictors of proactive coping. Reactive coping was positively predicted by the PTG dimension relating to others, depression, vitality, other than partner relation, and physical role, and negatively predicted by mental health level and emotional role. In summary, higher MH was associated with proactive coping strategies, whereas post-traumatic growth was related to the use of a wide range of proactive coping as well as reactive coping strategies.

## 1. Introduction

Multiple sclerosis (MS) is a chronic neurodegenerative disease frequently beginning in young adulthood. Myelin sheaths covering nerve fibers are attacked by the immune system. The result is inflammation that can lead to lesions in the brain, optic nerve, and spinal cord. Thus, communication between the central nervous system and other parts of the body is impaired [1].

The prevalence of MS is heterogeneous. The highest estimated prevalence is in Europe and America, exceeding 100 cases per 100,000 individuals, whereas the estimated prevalence in Africa and Southeast Asia is below 10 cases per 100,000 individuals. The number of people living with MS has increased significantly in recent decades, with a female predominance in the incidence of the disease worldwide [2]. Symptomatology associated with MS includes fatigue, chronic pain, balance problems, spasticity, cognitive impairment, visual disturbances, and bowel problems. In socio-economic terms, MS patients have been found to have 15–30% lower employment and earnings, as well as higher absenteeism and work disability [3]. A significant proportion of MS patients, therefore, require informal care from family or friends [4]. Informal caregivers may experience distress and reduced well-being affecting physical and mental health (MH) [5,6,7]. All participants in a study by Gafari and colleagues [8] suffered at some point from emotional symptoms, including anxiety, depression, and stress. Participants identified hard and extended periods of care as the origin of their disturbance. Hlabangana and Hearn [6] reported that half of the caregivers in their sample fulfilled the criteria for depression. They also scored lower on all quality of life domains compared to the general population.

Although caregiving repercussions have been widely reported, MS care can also lead to beneficial consequences, for instance, becoming aware of personal resources and family support [4,9]. Thus, complications in MS caring may coexist with positive outcomes, such as benefit finding or post-traumatic growth (PTG). PTG is generally defined as an experience of positive change that occurs as a result of the struggle with highly challenging life crises. “Post-traumatic” refers to the growth that occurs after a crucial adverse event in life. “Growth” denotes a personal enhancement of capabilities and functioning due to a superior mental and emotional awareness resulting from a critical life [10]. Regarding PTG’s connection with MH, PTG has been shown to be positively related to anxiety and negatively related to depressive symptoms. The lack of energy from depression can hinder the inner drive needed for growth [10], while anxiety might leave the subject in a state more favorable to respond actively to adversity. 

In the process of adapting to the caring role and struggling with its duties, the use of a wide spectrum of coping strategies has been emphasized. Early research studies in this field emphasized that in some cases, the efforts of caregivers and patients, due to conflicting coping styles, may result in even more stress in an already stressful situation [11]. Different coping styles shape the background of diverging types of communication when dealing with MS, such as talking about the illness or not and communicating emotions or not [12]. An interesting finding is the observed gender-related use of coping styles; for example, men tend to use more planning as a coping strategy, while women tend to designate their own physical space within the house [13]. As far as the typical coping styles in MS caregivers are concerned, these have been investigated in order to identify the specific ways caregivers respond to stress and demand, but no reasonable conclusions have been reached yet, probably because coping is a highly complex phenomenon [14]. In this context, Lazarus and Folkman’s Transactional Model of Stress and Coping [15] may help to comprehend the connection between caregivers’ roles and well-being. According to the model, caregiving consists of functional and emotional demands. Coping should keep caregiving demands and caregivers’ resources in balance to preserve their well-being [16]. There is evidence that stress management coping, health-promoting behaviors, supporting engagement, and positive reframing are helpful strategies for caregivers, contrary to avoidance coping [4,9]. Potentially, proactive coping strategies, which aim at controlling a situation in advance, are thought to be more adaptive compared to reactive strategies, which respond to a challenging situation [17]. Impaired physical and MH in caregivers have been proven to be an obstacle in caregiving, clarifying the need for adaptive coping strategies [4,5,8,18,19]. Various studies point out the beneficial effect of social support to preserve MS caregivers’ well-being and MH [9,20,21,22].

Against this backdrop, the thorough investigation of factors influencing the use of specific coping strategies is of great importance [5]. A deeper insight into the use of coping strategies in MS caregivers facilitates the understanding of their specific needs. In the long run, it may help to tailor interventions to reduce caregiver strain and burden to corroborate the quality of life and reduce poor outcomes among both caregivers and care recipients with MS. A recent comprehensive review on caregiving in MS comes to the conclusion that current knowledge on MS caregiving is limited particularly with regard to approaches remediating caregiver burden [4]. 

The present study is aimed to explore the use of different coping strategies in MS caregivers with special emphasis on PTG and MH levels and identify potential biopsychosocial predictors of proactive and reactive coping.

## 2. Materials and Methods

### 2.1. Sample and Procedure

Principal caregivers of MS outpatients from Virgen Macarena University Hospital in Seville, Spain, were asked to take part in the study from June 2017 to May 2018. 

The inclusion criteria were as follows: (a) being the caregiver of a patient with a confirmed MS diagnosis; (b) being over 18; and (c) being able to understand and fill in study questionnaires. The participants answered the questionnaires themselves. The research psychologist was available for any questions or assistance required. Figure 1 summarizes the sample selection process.

The research was authorized by the responsible Ethics Committee (0846-N-18). All participants gave their informed consent to participate in the study.

### 2.2. Instruments

#### 2.2.1. Health-Related Quality of Life

The 12-Item Short Form Health Survey (SF-12) consists of 12 items. The SF-12 includes eight domains: physical functioning, role-physical, bodily pain, general health, vitality, social functioning, role-emotional, and MH. Subscales scores vary from 0 (worst) to 100 (best). These subscales can result in two summary component scores: the Physical Component Summary Score (PCS) and the Mental Component Summary Score (MCS). PCS and MCS were calculated using QualityMetric SF Health Outcomes Scoring Software [23,24]. In our sample, Cronbach’s alpha for the mentioned dimensions ranged from 0.89 to 0.92. Cronbach’s alpha for PCS and MCS were 0.92 and 0.88, respectively [25].

#### 2.2.2. Mental Health Questionnaire (GHQ-28)

The General Health Questionnaire (GHQ-28) contains 28 items on a 4-point Likert scale. The four subscales are somatic symptoms, anxiety/insomnia, social dysfunction, and depression. Subscale scores range from 0 (best) to 21 (worst). The total GHQ-28 score ranges from 0 (best) to 84 (worst) [26,27]. The GHQ-28 Spanish version presents an acceptable degree of validity [28] and reliability in the investigation of chronic medical conditions [29]. In our sample, Cronbach’s alpha for the subscales varied from 0.77 to 0.92. Cronbach’s alpha for the total scale was 0.92.

#### 2.2.3. Post-Traumatic Growth

Perception of personal gain after MS was measured using the Spanish version of the Post-Traumatic Growth Inventory (PGI-21) [30,31]. The PGI-21 contains 21 items scored on a 5-point Likert scale, with a higher score indicating greater change. Test results provide five dimensions: relating to others, new possibilities, personal strength, spiritual change, and appreciation of life. Cronbach’s alpha in our study ranged from 0.77 to 0.88 for the subscales and 0.94 for the total score.

#### 2.2.4. Coping Strategies

The Spanish version of the Brief COPE Questionnaire (COPE-28) assessed caregivers’ coping strategies [32,33]. The COPE-28 comprises 28 items scored on a 4-point Likert scale from 0 (“I have not been doing this at all”) to 3 (“I have been doing this a lot”). Items can be collapsed into 14 subscales: (1) acceptance; (2) emotional support; (3) humor; (4) positive reframing; (5) religion; (6) active coping; (7) instrumental support; (8) planning; (9) behavioral disengagement; (10) denial; (11) self-distraction; (12) self-blaming; (13) substance use; and (14) venting. Cronbach’s alpha in our study varied from 0.65 to 0.84 for the subscales.

#### 2.2.5. Social Support

The Multidimensional Scale of Perceived Social Support (MSPSS) evaluated perceived social support from family, friends, and partners or significant others [34,35]. The MSPSS consists of 12 items scored on a Likert scale from 1 to 7. The total score ranges from 12 to 84. In the present sample, Cronbach’s alpha varied from 0.91 to 0.95 for the subscales and 0.92 for the total score.

### 2.3. Data Analysis

Sample features were reported using descriptive analysis. A one-way analysis of variance (ANOVA) for quantitative variables (age, Expanded Disability Status Scale –EDSS–, months since diagnosis, and months since the outbreak) and a Chi-squared test for categorical variables (gender, partnership, occupation, educational level, MS subtype, family relation, and living together) explored possible differences in demographics and patients’ clinical characteristics between subgroups with low, medium, or high PTG or MH levels.

A 3 × 3 factorial ANOVA with Bonferroni post hoc tests was calculated to study the influence of post-traumatic growth level and MH level on the use of coping strategies.

Stepwise regression was used to determine biopsychosocial predictors of coping strategies. We distinguished proactive from reactive coping approaches [17]. Proactive coping is characterized by aiming at controlling the situation rather than acting in advance to tackle it and subsuming the following strategies: acceptance, emotional support, humor, positive reframing, religion, active coping, instrumental support, and planning. In the second coping approach, cognitive or emotional reactions, mostly resulting in avoidance, are at the forefront. Reactive strategies embraced behavioral disengagement, denial, self-distraction, self-blaming, substance use, and venting. The two coping styles were dependent variables in the multivariate model. Demographics (age, gender, partnership, occupation, educational level, family relation, and living together), clinical characteristics (EDSS, MS subtype, months since diagnosis, and months since the outbreak), SF-12, GHQ-28, PTG-21, and MSPSS subscales were introduced as predictors.

Statistics were computed using SPSS Statistics version 26. The significance level was set to *p* < 0.05. G*Power 3.1 Software calculated effect size coefficients. Coefficients were interpreted according to Cohen’s recommendations: for w (0.10 = small, 0.30 = medium, and 0.50 = large), for f (0.10 = small, 0.25 = medium, and 0.40 = large), f^2^ (0.02 = small, 0.15 = medium, and 0.35 = large effects), and d (0.20 = small, 0.50 = medium, and 0.80 = large effects) [36].

## 3. Results

The sample comprised 209 MS principal informal caregivers, 111 (53.08%) women and 98 (46.92%) men. The mean age was 47.5 (SD = 13.44). Table 1 and Table 2 present sample demographics and MS patients’ clinical features. The family relationship between caregiver and patient was as follows: partner (64.6%), parent (17.2%), child (9.1%), sibling (6.2%), and other (2.9%).

### 3.1. Influence of Post-Traumatic Growth and Mental Health on Coping Strategies

The influence of post-traumatic growth and MH on the use of coping strategies was explored.

The 209 participants were divided into three groups according to their PGI-21 score: 69 participants with low PTG (33.01%; 0–37 points), 69 participants with medium PTG (33.01%; 38–60 points), and 71 participants with high PTG (33.98%; 61–101 points). Furthermore, participants were divided into three groups based on their MH level as measured by the GHQ-28 score: 66 participants with high MH and low GHQ-28 score (31.58%; 4–13 points), 73 participants with medium MH (34.92%; 14–21 points), and 70 participants with low MH (33.5%; 22–72 points).

There was a statistically significant difference in gender (*p* < 0.0001) between the three MH groups, with more men in the low MH group and women in the high MH group with a medium effect size (Table 2). Results did not show any interaction effect between PTG and MH on coping strategies.

Main effects showed a significant influence of PTG on the following coping strategies: emotional support [F_(2, 200)_ = 13.691, *p* < 0.0001], positive reframing [F_(2, 200)_ = 8.035, *p* < 0.0001], religion [F_(2, 200)_ = 13.671, *p* < 0.0001], active coping [F_(2, 200)_ = 5.506, *p* = 0.005], instrumental support [F_(2, 200)_ = 5.306, *p* = 0.006], planning [F_(2, 200)_ = 3.267, *p* = 0.040], denial [F_(2, 200)_ = 3.361, *p* = 0.037], self-distraction [F_(2, 200)_ = 5.339, *p* < 0.0001], self-blaming [F_(2, 200)_ = 9.636, *p* < 0.0001], and venting [F_(2, 200)_ = 3.554, *p* = 0.030]. Effect size coefficients f varied from 0.176 to 0.369, pointing to small to medium effects (Table 3). Between groups, the comparison showed the above-mentioned. The comparison of patients’ and caregivers’ PTG1-21 subscale scores at T1 did not show statistically significant differences. Mean scores, standard deviation, and paired t-test results are reported in Table 4.

The MH factor was significant for acceptance [F_(2, 200)_ = 4.539, *p* = 0.011], behavioral disengagement [F_(2, 200)_ = 4.023, *p* = 0.019], and self-distraction [F_(2, 200)_ = 3.521, *p* = 0.031], with small effect sizes f (0.188–0.215) (Table 3). As presented in Table 5, acceptance was more frequently used by the high MH group, while behavioral disengagement and self-distraction were more frequently used by the low MH group. Respective associations showed small effect sizes (d varies from 0.403 to −0.494).

### 3.2. Coping Strategies Predictors

Proactive and reactive coping were regressed on demographics, clinical characteristics, SF-12, GHQ-28, PGI-21, and MSPSS.

Greater proactive coping was predicted in descending order of contribution by relating to others (β = 0.263, *p* = 0.002), not living together with the patient (β = 0.244, *p* < 0.001), partner or significant other’s social support (β = 0.171, *p* < 0.003), new possibilities (β = 0.186, *p* = 0.022), physical role (β = 0.257, *p* = 0.001), and emotional role (β = 0.223, *p* = 0.004). These six variables accounted for 34.6% of proactive coping variance with a large effect size (Table 6).

Reactive coping was positively predicted by relating to others (β = 0.327, *p* < 0.001), severe depression (β = 0.228, *p* < 0.001), vitality (β = 0.293, *p* < 0.001), other than partner relation (β = 0.134, *p* < 0.013), and physical role (β = 0.158, *p* < 0.001). Reactive coping was negatively predicted by MH (β = −0.289, *p* < 0.001) and emotional role (*β* = −0.241, *p* = 0.002). All variables introduced in the model explained 46.3% of the variance. The effect size coefficient indicated a large effect size (Table 6).

## 4. Discussion

MS caregivers’ coping has shown to be a key factor in the patient–caregiver dyad’s adaptation to the disease and well-being. The present study explored the influence of PTG and MH levels on the use of different coping strategies as well as biopsychosocial predictors of proactive and reactive coping.

### 4.1. Influence of Post-Traumatic Growth and Mental Health on Coping Strategies

No interaction effect was found between PTG or MH and coping strategies. 

Regarding the main effects, post-traumatic growth showed a significant effect on coping. Caregivers with higher PTG used more frequent coping strategies such as emotional support, positive reframing, religion, active coping, instrumental support, planning, denial, self-distraction, self-blaming, and venting. 

In reference to emotional and instrumental support, our results are in line with previous research pointing out the connection between opening up in a secure social environment and trauma processing [37]. The promotion of communication and connection between MS family members is related to disease adaptation and higher PTG [21,38]. Asking for practical help involves perceiving one’s own needs and expressing the emotional stress associated with it [37].

Positive reframing has been extensively related to benefit and personal gain as post-traumatic growth [37,39,40]. This strategy implies a positive cognitive reconstruction of a challenging situation. Interventions focused on re-evaluating goals and values have been demonstrated to support meaningfulness and well-being in caregivers [41]. In the same direction, religious coping has been associated with PTG by re-evaluating despairing situations into challenging assignments. Moreover, religious communities can enhance relationships via participation in communal activities [37,40].

From the caregivers’ perspective, planning and confronting themselves with the fact of an unpredictable disease progression may increase a feeling of being in control and being able to pursue new goals, which may explain the association with PTG in our study. 

Even though denial and self-distraction are regarded as disadvantageous coping strategies in many circumstances, our results agree with earlier research arguing a potentially positive relationship between avoidance coping and personal growth in MS carers [18,42,43]. The capability to transiently divert one’s attention from an uncontrollable situation might be helpful in maintaining emotional stability [44]. Pakenham [42] distinguished between two types of avoidance or withdrawal: First, complete flight from stress, burying the head in the sand, resulting in distress and maladjustment. Second, transient “time out” to reconsider priorities, gather strengths, and plan, which can be related to personal growth and gain.

Very little was found in the literature on self-blaming and PTG. Bearing in mind that post-traumatic growth, as conceptualized by Calhoun and Tedeschi [45], derives from an intense inner confrontation with challenging situations, it involves self-reflection and at least transient self-criticism that puts the handling of previous situations and the application of familiar strategies into question. Thus, self-blaming could be understood in this context as a transient strategy fueling inner cognitive and emotional reorganization.

Concerning venting, COPE-28 conceptualized this coping strategy as an emotional reaction to liberate oneself from negative emotions. To accomplish venting and benefit from its cathartic effect, caregivers need to recognize unpleasant emotions and allow themselves to show their feelings [42].

MH did also have an impact on caregivers’ coping. Better MH was related to acceptance, while behavioral disengagement and self-distraction were associated with worse MH. These results are supported by previous research by Penwell-Waines et al. [19], who found that caregivers with lower levels of distress employed more self-care and stress management. The findings of Bassi et al. [16] associated caregivers’ avoidance with poorer well-being. As outlined above, on the one hand, the ability to transiently distract from a challenging task might be beneficial to spare resources. Thus, O’Brien [11] reported that MS caregivers regularly used wishful thinking to modify the current situation and distract themselves from the harsh reality. However, in poor MH associated, for example, with depression or anxiety, self-distraction and avoidance may become the predominant strategies, resulting in maladaptation and worsening of the situation. Obviously, the COPE-28 questionnaire does not distinguish between the transient or predominant use of specific strategies.

### 4.2. Coping Strategies Predictors

The first predictor of proactive coping defined in our study was relating to others. This PGI-21 subscale appraises closeness to others, compassion for those who suffer, and the willingness to be helped and use social support. Caregivers who rely on their social network instead solve problems in conjunction with others and communicate their personal experiences [11]. Seeking emotional and instrumental support, and positive reframing, might be easier for those caregivers who have a supportive social environment. A lot of evidence connects caregivers’ adaptation and healthier coping with close family relationships and social support [9,20,21,22].

Not living together with the patient predicted higher use of proactive coping. Caregivers not living apart may be overtaxed due to long periods of care without being able to distance themselves from the challenging task and take care of themselves. In most cases, if the principal caregiver is not living with the patient, this indicates a lower level of disability, not requiring assistance 24 h daily [6]. 

Not surprisingly, partner support was a positive predictor of proactive coping. Most study participants were in a partnership with the patient (64.6%). Couples’ satisfaction has been demonstrated to play an important role in the patient–caregiver dyad’s adjustment. Prior research stated that couples reporting a high-quality partnership cope better with emotional stress [21,46,47]. In fact, intervention programs incorporating partner support in MS have been shown to improve adaptation in both members [7,38,48].

The PGI-21 dimension of new possibilities was also a positive predictor of proactive coping. It can be argued that meeting new people and engaging in new activities or life projects is closely associated with a tendency to tackle challenging situations in advance rather than merely reacting to them [37]. In the same direction, fewer physical and emotional limitations as measured by the SF-12 scales role-physical and role-emotional predict proactive coping as less health impairment facilitates handling and controlling a problematic situation. The relationship between greater well-being and proactive coping has been proven in previous studies [4,16].

Unexpectedly, relating to others was also the strongest predictor of reactive coping. One might argue that social contacts and relations can also be used to self-distract oneself from an unbearable situation, or their help may lead to behavioral disengagement. However, as outlined above, this reactive coping strategy, at least when transiently used, may also be adaptive as they help to spare resources and gain strength. 

Higher depression predicted more frequent reactive coping, whereas better MH predicted less. These findings are broadly supported by previous research connecting reactive coping with a higher probability of maladaptation to the caregiving situation [16,42]. Improvement of depression and distress led to more health-promoting and stress management behavior [18,19], highlighting the relevance of stress prevention in MS caregiving.

Another family relationship between the caregiver and partner also predicted reactive coping. Several reports have shown that a “communal sense of the illness” can foster positive outcomes in the MS patient–caregiver dyad [38,46,48]. Developing a communal sense of MS requires intense mutual communication involving patients’ and caregivers’ inner perspectives, which may be easier in a partnership [49].

Contrary to expectations, vitality predicted higher use of reactive coping. One might argue that vitality may be associated with the intense wish to take part in social life and associated activities. If this desire is hampered by a particularly challenging caregiving situation reactive strategy, self-distraction or denial may be necessary and at least transiently more frequently used.

The major study weakness is the non-random selection of the sample, which limits its external validity. Additionally, only self-report instruments were applied, and there are scarce data on the validation of use in the population of MS caregivers. Furthermore, some factors should have been controlled, such as age, gender, or time since diagnosis, to avoid some possible interactions in the results. Considering certain variables, such as the level of dependency of the MS patient on the caregiver, would have been very appropriate. 

Its major strength is the large and heterogeneous sample and the investigation of a variety of biopsychosocial variables.

## 5. Conclusions

The findings of this study have high relevance for the preservation and improvement of the well-being of MS caregivers in the clinical practice and, thus, for the care of patients. MH and post-traumatic growth have an important impact on the use of coping strategies in MS caregivers. Poor MH is associated with less proactive coping, whereas post-traumatic growth corroborates the use of a wide range of proactive coping as well as reactive coping strategies. The results highlight the importance of multidisciplinary interventions to preserve MH and enhance those strategies that are most adaptive for each caregiver.

Self-distraction and other reactive coping strategies may at least transiently be adaptive in caregivers. Preventive measures regarding MH and support of post-traumatic growth may increase the probability of more frequent proactive coping and increase the probability of favorable and adaptive coping in the long run. In this line, Acceptance and Commitment Therapy (ACT) is intended to help in accepting uncomfortable feelings instead of eliminating them. Caregivers could obtain a more flexible approach to unpleasant feelings and avoid maladaptive coping [50]. Moreover, supporting social environments and connections should be considered a priority to facilitate the adaptation in MS caregivers. A variety of therapies, including social skills training and self-help groups, are proven to promote the usage of social supports by promoting communication, strengthening social bonds, and facilitating sharing of emotions in a safe social environment [49,51]. Future research is recommended to longitudinally explore the impact of biopsychosocial factors on MS caregivers’ coping. 

## Figures and Tables

**Figure 1 healthcare-11-01390-f001:**
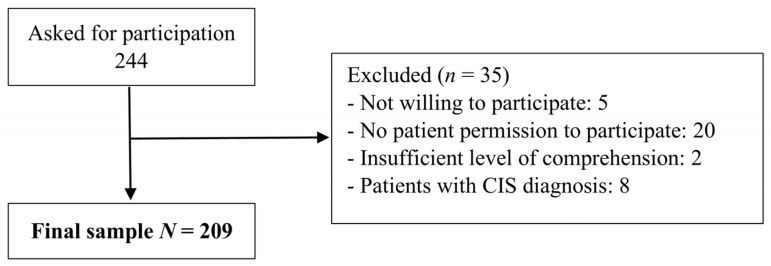
Study flowchart. CIS = Clinically isolated syndrome.

**Table 1 healthcare-11-01390-t001:** Comparison of sociodemographic and clinical characteristics of the three different post-traumatic growth level groups.

	Post-Traumatic Growth Level			Intergroup Comparison		Effect Size
	Low (*n* = 69)	Medium (*n* = 69)	High (*n* = 71)	χ^2^	*p*	Cohen’s w
Gender n (%)				1.67	0.433	(N)
Male	28 (40.57)	35 (50.72)	35 (49.29)			
Female	41 (59.43)	34 (49.28)	36 (50.70)			
Partnership n (%)				1.104	0.576	(N)
No partner	12 (17.39)	8 (11.59)	12 (16.90)			
Partner	57 (82.61)	61 (88.41)	59 (83.09)			
Occupation n (%)				7.569	0.053	(S)
Employed/In education	33 (47.82)	48 (69.56)	46 (64.78)			
Unemployed	36 (52.18)	21 (30.44)	25 (35.22)			
Educational level n (%)				9.449	0.051	(S)
Primary education	24 (34.78)	11 (15.94)	19 (26.76)			
Secondary education	24 (34.78)	21 (30.43)	23 (32.39)			
University or higher	21 (30.44)	37 (53.63)	29 (40.85)			
MS subtype n (%)				3.630	0.163	(N)
Remittent	50 (7.25)	57 (82.61)	60 (84.51)			
Progressive	19 (92.75)	12 (17.39)	11 (15.49)			
Family Relation				4.087	0.130	(N)
Partners	38 (55.07)	48 (69.57)	49 (69)			
Others	31 (44.93)	21 (30.43)	22 (31)			
Cohabitation				2.333	0.311	(N)
Yes	50 (72.46)	51 (73.91)	59 (83.10)			
No	19 (27.54)	16 (26.09)	12 (16.90)			
				F _(2, 206)_	*p*	Cohen’s d
Age (M ± SD)	49.97 ± 13.19	44.64 ± 12.46	47.87 ± 14.13	2.817	0.062	(S)
EDSS (M ± SD)	3.36 ± 2.08	3.84 ± 2.07	3.67 ± 2.34	1.046	0.353	(N)
Months since diagnosis (M ± SD)	153.96 ± 97.41	1.34.49 ± 89.16	158.23 ± 88.29	1.324	0.268	(N)
Months since outbreak (M ± SD)	195.16 ± 114.13	165.99 ± 102.33	201.76 ± 120.87	1.981	0.141	(N)

N = null effect size; S = small effect size; MS = Multiple sclerosis; EDSS = Expanded Disability Status Scale.

**Table 2 healthcare-11-01390-t002:** Comparison of sociodemographic and clinical characteristics of the three different mental health level groups.

	Mental Health Level			Intergroup Comparison		Effect Size
	Low (*n* = 66)	Medium (*n* = 73)	High (*n* = 70)	χ^2^	*p*	Cohen’s w
Gender n (%)				17.359	<0.0001	(M)
Male	44 (66.67)	32 (43.84)	22 (31.43)			
Female	22 (33.33)	41 (56.16)	48 (68.57)			
Partnership n (%)				0.775	0.679	(N)
No partner	11 (16.67)	9 (12.33)	12 (17.14)			
Partner	55 (83.33)	64 (87.67)	58 (82.86)			
Occupation n (%)				1.862	0.397	(N)
Employed/In education	42 (63.64)	47 (64.38)	38 (54.29)			
Unemployed	24 (36.36)	26 (35.62)	32 (45.71)			
Educational level n (%)				5.265	0.261	(N)
Primary education	16 (24.24)	17 (23.30)	21 (30)			
Secondary education	16 (24.24)	28 (38.35)	24 (34.29)			
University or higher	34 (51.52)	28 (38.35)	25 (35.71)			
MS subtype n (%)				0.090	0.956	(N)
Remittent	52 (78.79)	59 (80.82)	56 (80)			
Progressive	14 (21.21)	14 (19.18)	14 (20)			
Family Relation				3.099	0.212	(N)
Partners	48 (72.73)	46 (63)	41 (58.57)			
Others	18 (27.27)	27 (37)	29 (41.43)			
Cohabitation				0.498	0.780	(N)
Yes	53 (80.30)	55 (75.34)	54 (77.14)			
No	13 (19.70)	18 (24.66)	16 (22.86)			
				F_(2, 206)_	*p*	Cohen’s d
Age (M ± SD)	50.39 ± 12.64	47.21 ± 13.33	45.03 ± 13.83	2.794	0.063	(S)
EDSS (M ± SD)	3.69 ± 2.32	3.68 ± 2.14	3.67 ± 2.11	0.491	0.613	(N)
Months since diagnosis (M ± SD)	151.23 ± 71.21	140.70 ± 86.09	155.50 ± 113.07	0.400	0.400	(N)
Months since outbreak (M ± SD)	184.82 ± 88.94	181.15 ± 116.23	197.46 ± 130.42	0.001	0.999	(N)

N = null effect size; S = small effect size; M = medium effect size; MS = Multiple sclerosis; EDSS = Expanded Disability Status Scale.

**Table 3 healthcare-11-01390-t003:** Coping strategies: differences in coping strategies used by post-traumatic growth and mental health (3 × 3 factorial analysis of variance).

	Main Effects (Cohen’s f)		Interactive Effects
COPE-28	Post-Traumatic Growth	Mental Health	F _(2, 200)_ *p*
	F _(2, 200)_	F _(2, 200)_	
Acceptance	2.410 0.157 S	4.593 * 0.215 S	0.463 0.763
Emotional support	13.691 ** 0.369 M	2.616 0.160 S	0.327 0.859
Humor	2.531 0.160 S	0.188 0.044 N	0.558 0.693
Positive reframing	8.035 ** 0.282 M	0.778 0.089 N	0.514 0.726
Religion	13.671 ** 0.368 M	0.957 0.095 N	1.087 0.364
Active coping	5.506 * 0.234 S	0.382 0.063 N	0.239 0.916
Instrumental support	5.306 * 0.229 S	0.006 0.031 N	0.215 0.930
Planning	3.267 * 0.181 S	0.424 0.063 N	0.545 0.703
Behavioral disengagement	3.069 0.175 S	4.023 * 0.201 S	0.871 0.482
Denial	3.361 * 0.176 S	3.128 0.185 S	1.224 0.302
Self-distraction	5.339 * 0.232 S	3.521 * 0.188 S	0.410 0.802
Self-blaming	9.636 ** 0.311 M	1.997 0.142 S	0.245 0.912
Substance use	2.040 0.142 S	1.038 0.100 S	1.402 0.235
Venting	3.554 * 0.188 S	2.363 0.153 S	1 0.409

N = null effect size; S = small effect size; M = medium effect size. Significance value * *p* < 0.05; ** *p* < 0.001.

**Table 4 healthcare-11-01390-t004:** Coping strategies: differences between coping strategies used according to post-traumatic growth level.

	Post-Traumatic Growth Level M (SD)			Comparisons *p* (Cohen’s d)		
	Low (a) *n* = 69	Medium (b) *n* = 69	High (c) *n* = 71	Group Levels		
				a-b	a-c	b-c
Acceptance	2.03 (0.72)	2.01 (0.63)	2.23 (0.74)	1 0.029 N	0.157 −0.274 S	0.201 −0.320 S
Emotional support	0.86 (0.68)	1.18 (0.76)	1.58 (0.94)	0.084 −0.443 S	<0.001 ** −0.877 L	0.009 * −0.468 S
Humor	0.81 (0.95)	0.81 (0.90)	1.12 (0.99)	1 0 N	0.135 −0.319 S	0.191 −0.327 S
Positive reframing	1.32 (0.74)	1.53 (0.77)	1.82 (0.78)	0.290 −0.278 S	<0.001 ** −0.658 M	0.062 −0.374 S
Religion	0.59 (0.83)	0.56 (0.80)	1.27 (1.15)	1 0.037 N	<0.001 ** −0.678 M	<0.001 ** −0.716 M
Active coping	1.78 (0.75)	1.95 (0.61)	2.16 (0.65)	0.394 −0.249 S	0.003 * −0.541 M	0.216 −0.333 S
Instrumental support	1.04 (0.70)	1.19 (0.63)	1.45 (0.83)	0.681 −0.225 S	0.004 * −0.534 M	0.134 −0.353 M
Planning	1.40 (0.82)	1.48 (0.70)	1.71 (0.84)	1 −0.577 M	0.042 * −0.807 L	0.237 −0.297 S
Behavioral disengagement	0.22 (0.47)	0.23 (0.42)	0.44 (0.66)	1 −0.022 N	0.164 −0.384 S	0.067 −0.377 S
Denial	0.25 (0.46)	0.47 (0.58)	0.51 (0.73)	0.158 −0.420 S	0.044 * −0.426 S	0.158 −0.061 N
Self-distraction	0.96 (0.72)	1.36 (0.85)	1.42 (0.83)	0.023 * −0.508 M	0.010 * −0.592 M	0.023 * −0.071 N
Self-blaming	0.52 (0.57)	0.79 (0.73)	1.14 (0.92)	0.176 −0.412 S	<0.001 * −0.810 L	0.041 * −0.421 S
Substance use	0.06 (0.24)	0.09 (0.27)	0.21 (0.58)	1 −0.117 N	0.456 −0.337 S	1 −0.265 S
Venting	0.63 (0.684)	0.75 (0.59)	0.97 (0.70)	1 −0.188 N	0.034 * −0.491 S	0.162 −0.339 S

N = null effect size; S = small effect size; M = medium effect size; L = large effect size. Significance value * *p* < 0.05; ** *p* < 0.001.

**Table 5 healthcare-11-01390-t005:** Coping strategies: differences between coping strategies used according to mental health.

	Mental Health Level M (SD)			Comparisons *p* (Cohen’s d)		
	Low (a) *n* = 66	Medium (b) *n* = 73	High (c) *n* = 70	Group Levels		
				a-b	a-c	b-c
Acceptance	2.28 (0.63)	2 (0.64)	1.99 (0.80)	0.043 * 0.441 S	0.018 * 0.403 S	1 0.013 N
Emotional support	1 (0.80)	1.31 (0.82)	1.30 (0.90)	0.073 −0.383 S	0.472 −0.352 S	1 0.011 N
Humor	0.95 (0.96)	0.86 (0.88)	0.94 (1)	1 0.097 N	1 0.010 N	1 −0.084 N
Positive reframing	1.63 (0.77)	1.49 (0.71)	1.55 (0.88)	0.817 0.189 N	0.135 0.096 N	0.132 −0.075 N
Religion	0.88 (1.11)	0.80 (0.95)	0.75 (0.94)	1 0.077 N	0.583 0.126 N	0.865 0.053 N
Active coping	1.96 (0.71)	1.99 (0.56)	1.96 (0.78)	1 −0.046 N	1 0 N	1 0.044 N
Instrumental support	1.21 (0.73)	1.20 (0.63)	1.27 (0.85)	1 0.015 N	1 −0.075 N	1 −0.093 N
Planning	1.58 (0.83)	1.53 (0.66)	1.51 (0.90)	1 0.066 N	1 0.080 N	1 0.025 N
Behavioral disengagement	0.18 (0.50)	0.27 (0.47)	0.44 (0.59)	0.934 −0.186 S	0.017 * −0.475 S	0.200 −0.318 S
Denial	0.25 (0.46)	0.46 (0.66)	0.50 (0.66)	0.117 −0.369 S	0.074 −0.439 S	1 −0.060 N
Self-distraction	1.04 (0.87)	1.21 (0.68)	1.47 (0.87)	0.708 −0.217 S	0.026 * −0.494 S	0.392 −0.332 S
Self-blaming	0.71 (0.84)	0.73 (0.69)	1 (0.83)	1 −0.026 N	0.247 −0.347 S	0.258 −0.353 S
Substance use	0.06 (0.27)	0.10 (0.34)	0.19 (0.53)	1 −0.130 N	0.456 −0.309 S	1 −0.202 S
Venting	0.64 (0.59)	0.78 (0.63)	0.92 (0.76)	0.630 −0.229 S	0.94 −0.411 S	1 −0.200 S

N = null effect size; S = small effect size. Significance value * *p* < 0.05.

**Table 6 healthcare-11-01390-t006:** Proactive and reactive coping strategies multiple linear regression models.

	F	R^2^	B	SE.B	β	1-β	f^2^
Proactive coping strategies							
Model	17.831 _(6, 202)_	0.346	0.203	0.189		0.99	0.529 (L)
Relating to others			0.102	0.032	0.263 *		
Cohabitation			0.289	0.070	0.244 **		
Partner or other support			0.062	0.021	0.171 *		
New possibilities			0.076	0.033	0.186 *		
Physical role			0.005	0.002	0.257 *		
Emotional role			0.004	0.002	0.223 *		
Reactive coping strategies							
Model	24.718 _(7, 201)_	0.463	0.319	0.119		0.99	0.862 (L)
Relating to others			0.097	0.016	0.327 **		
Severe depression			0.029	0.007	0.228 **		
Mental health			−0.006	0.001	−0.289 **		
Vitality			0.005	0.001	0.293 **		
Emotional role			−0.004	0.001	−0.241 *		
Other than partner relation			0.107	0.043	0.134 *		
Physical role			0.002	0.001	0.158 *		

L = large effect size. Significance value * *p* < 0.05; ** *p* < 0.001.

## Data Availability

Data Availability Statements can be found at https://www.mdpi.com/ethics. The raw data backing the conclusions of this investigation will be made available by the authors upon demand. Accessed on 3 March 2023.

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
