# Peer review of "Significance of Post-Traumatic Growth and Mental Health for Coping in Multiple Sclerosis Caregivers"

_healthcare, 2023, doi:10.3390/healthcare11101390_

Round 1
Reviewer 1 Report
This paper investigates the impact of post-traumatic growth (PTG) and mental health(MH) on the use of coping strategies by caregivers with multiple sclerosis (MS) and identifies predictors of biopsychosocial active or passive measures. This thesis assessed 209 caregivers through a questionnaire approach. Results found that higher PTG was associated with greater use of emotional support, positive reframing, religion, active coping, instrumental support planning, denial, self-transference, self-blame, and venting. Better psychological well-being was associated with greater use of acceptance, while behavioral disengagement and self-transference were associated with poorer psychological well-being. ln conclusion, higher levels of psychological well-being were associated with positive coping strategies, whereas post-training traumatic growth was associated with the use of a wide range of positive and reactive coping strategies. However, the following questions need to be responded to and addressed prior to the publication of this paper.
First of all, there are the following points to note and consider in terms of the format of the article. (1) In terms of explanation of some proper nouns and abbreviation of proper nouns, authors should check again throughout the text for writing errors, and at the same time, authors should check again throughout the text for hyphenation and errors inexpression logic, and if so, please identify and correct them; (2) The format of references should be standardized, both in the text and in the appendices. Authors are requested to check the format of references again according to the requirements of the journal; (3) lt is also necessary for authors to check and correct some formatting errors in the format of pictures and tables in the text again.
Second, in terms of the content of the article. (1) The introduction emphasizes the positive effects of proactive adaptive measures on both caregivers and caregivers, and cites some literature to support this view. However, we need to pay special attention to the fact that the selection of the literature needs to focus on the continuity of the content and the continuity of the time. The introduction of this article is mainly a brief development of the background of the study and does not lay out the chronological sequence of the previous scholars' views on the topic of "the emphasis on the use of a wide range of coping strategies in adapting to and struggling with the caregiving role "ln the last paragraph of the introduction, there is too little discussion of the importance and necessity of this study. lt is suggested that this section could be enriched by adding some supporting literature and expanding the statement of the necessity and importance of this study. (2) ln the section on sample and data, the authors state the source of the sample data, the scientific validity of the scale, and the conformity and credibility of the study. I think this study deserves recognition that the use of operationalized scales and data analysis software in applied psychology is very good, which can fully reflect the author's professionalism. However, how did the authors ensure the credibility and validity of this data collection by ensuring that differences in factors such as the age and gender of the respondents would affect the collection of the questionnaire, for example, if the respondents were too old or too young, this would affect the collection of the questionnaire? in other words how did the authors set the control variables and how did they avoid possible "interactions" between the independent variables which may lead to errors in the results? (3) In Table 2 of the article, I was able to observe that the study was multidisciplinary in nature, i.e., sociology, applied psychology, and medicine, which is very good and closely aligned with the frontiers of research. (3) In terms of discussion and conclusions, this study explored the effects of PTG and MH levels on the use of different coping strategies and the biopsychosocial predictions of active and passive coping. lt should also be noted that the paper should be of value to service practice and the future value of this research should be fully known to the paper's creators or the patients targeted by this study. I suggest that the conclusion section could be supplemented with a description of the relevance of the study based on the analysis of the data. Also, it is necessary to add a logical and appropriate description of the limitations of the study to reflect the structural integrity of the paper.
Finally, I think this paper combines actual data, uses models and data analysis software. And draws conclusions with strong practical significance. expect the author to pass on three copies of the original scales involved in the article when he replies, showing the author’s professionalism, so that he can better obtain permission to publish the paper. Good luck with the publication of the paper.

Reviewer 2 Report
Why was Posttraumatic Growth and Mental Health selected in this manuscript from the perspective of Multiple Sclerosis?
What are the main scientific questions addressed by this study?
Needing to add some limitations to the study area analysis, the theoretical logical relationship of posttraumatic growth and mental health, the basis of the selected plots and sample, and targeted suggestions should be more clearly illustrated.
Reviewer 3 Report
Very interesting study with the use of colleagues
Reviewer 4 Report
The submitted manuscript is of high quality and I have no major comments to make. However, I have a few minor comments. They are as follows:
1. Please add information on who are the intended administrators of the instruments used (caregivers, patients, doctors...).
2. What is the timeframe for data collection?
3. Please explain any acronyms (see e.g. EDSS on line 119 and 138).
4. Table 6 is not positioned correctly, see "Note" position on line 215; paragraph on lines 217-222.
Reviewer 5 Report
First of all I must say that the subject is interesting.
There are issues to improve the article, which I will now mention:
In the introduction it would be important to better characterise the pathology, known incidence and prevalence rate as well as of those how many depend on others and are cared for by relatives.
It would also be interesting to understand if there are data on the socio-economic impact of this pathology both for the patient and the carers. This is to better substantiate and justify the relevance of the topic.
The authors should explain further the concept of post-traumatic growth (PTG).
In the inclusion criteria it does not specify the time of diagnosis or the minimum time as a caregiver. If it has not been done, it represents a limitation to be mentioned in the study.
The instruments should include information on the validation of each one for the population in question and the respective reference.
It is not in the article what EDSS means.
In the results it would be important to know the sociodemographic and clinical data of the sample in a separate table.
I did not find the level of dependency of the family member with MS in the sample characterisation. Was this information not collected? Wouldn't it be important? If not why and, again, it represents a limitation.
In the discussion when you state that "From the carers' perspective, planning for and confronting the fact of unpredictable disease progression can increase the sense of being in control and can therefore be reassuring", is it the results of this study that you draw on for this conclusion? Or on another source? it should be clear.
In the conclusion it will be important to mention the implications of these findings for clinical practice. What are their usefulness and importance.
More than 40% of the references are over 10 years old. It would be important to update the sources to make them more recent.
Round 2
Reviewer 1 Report
Accept in present form
Reviewer 5 Report
I consider the work to be of sufficient quality for publication.